# How to Choose the Suitable Steel of Wellhead, Wellbore, and Downhole Tools for Acid Gas Reinjection Flooding

Yudi Geng [1,*], Zhendong Liu [2,*], Wenguang Zeng [1], Yibo Feng [1], Baodong Ding [1], Wenxia Li [1] and Qinying Wang [3]

1   Petroleum Engineering Technology Research Institute, Sinopec Northwest Oilfield Company, Urumchi 830011, China
2   State Key Laboratory of Oil & Gas Reservoir Geology and Exploitation, Southwest Petroleum University, Chengdu 610500, China
3   Materials Science and Engineering, Southwest Petroleum University, Chengdu 610500, China
*   Correspondence: gengyd.xbsj@sinopec.com (Y.G.); liuzd0522@foxmail.com (Z.L.)

**Abstract:** The material selection of injection gas wells in acid gas flooding is the bottleneck of the successful implementation of the technical scheme. Through standard and literature research, the materials of the wellhead, wellbore, and packer for reinjection well in acid gas flooding are preliminarily established, and then the suitable materials are further screened by using the weight-loss and surface characterization method. Finally, a new type of packer is designed to optimize the wellbore material. The results show that 35CrMo ($C_R$ = 0.0589 mm/y) steel is used for wellhead materials, 625 alloy steel is selected as the sealing surface, and 625 or 825 alloys (with $C_R \leq 0.0055$ mm/y) steel is used for wellhead sealing material. The main material of the packer is 718 Alloy (with $C_R \leq 0.0021$ mm/y). The cost of T95 steel within 20 years (1263 ten thousand yuan) of service is much smaller than that of G3 alloy (1771 ten thousand yuan), but after 30 years of service, its cost is close to that of G3 alloy. A kind of downhole packer for acid gas reinjection is designed. Among them, G3 alloy steel tubing is used between the packer and the relief valve, T95 steel tubing is selected above the packer and below the safety valve, and the packer is set in the G3 steel tubing. The serious pitting corrosion of T95 steel in the liquid phase environment is due to the uneven deposition of FeS and $FeCO_3$ on the steel surface.

**Keywords:** acid gas reinjection flooding; material selection scheme; $CO_2$-$H_2S$ corrosion; pitting corrosion; packer device

## 1. Introduction

So far, acid gas reinjection flooding has been widely used in North American oilfields. Acid gas flooding has become an efficient means of achieving a double gain in economic development and environmental protection. For one thing, acid gases bring the greenhouse effect, acid rain, and environmental damage if they are directly discharged into the atmosphere. Another thing is the acid gases corrode pipelines and equipment, leading to the cost of acid gas treatment being extremely high. However, acid gas is injected directly into the formation, which can avoid greenhouse gases and reduce the cost of the purification plant [1–4]. In addition, sour gas is an effective displacement agent that can bring crude oil into the formation to the production well. In spite of all this, some problems remain to be solved in acid gas reinjection floods, in particular in the selection of oil and gas well materials.

It is crucial to inject acid gas into training safely because of the high corrosivity of $CO_2$ and $H_2S$ [5–7]. Researchers have also conducted thorough research on the corrosion of steel in the $CO_2$-$H_2S$ environment [8–11] and proposed lots of corrosion mechanism models [12–15]. Santos et al. noted that $FeCO_3$ formation was encouraged at high pH and 120 °C, while FeS structures were dominant at 90 °C [16]. Souza et al. also observed that protective corrosion conditions were reached at 120 °C with and without $H_2S$. Without $H_2S$, a stable and denser film of $FeCO_3$ was created. With $H_2S$, the FeS film decreased the



corrosive effect of $CO_2$ and delayed the precipitation of $FeCO_3$ [17]. Dong et al. explained the process of producing 3Cr steel corrosion product film in the $CO_2$-$H_2S$ environment. $Cr(OH)_3$ and FeS were competitively deposited on the surface of 3Cr steel. The FeS preferentially accumulates to form the inner layer and inhibit precipitation of $Cr(OH)_3$, whereas $Cr(OH)_3$ deposits away from the surface to form the outer layer [18]. However, the tubing is faced with high temperature and high pressure and the total pressure even exceeds 50 MPa due to the fact that the acid gas is compressed in acid reinjection. Unfortunately, it is rarely reported that $CO_2$ and $H_2S$ corrode tubing in such severe environments. Therefore, it is necessary to test the corrosion of tubing in such a harsh environment to understand the corrosion risk of tubing during operation.

In addition, the material selection of well facilities, such as a wellhead, wellbore, and packers, also needs further exploration. Dong et al. analyzed the main influencing factors of tubing corrosion in $CO_2$ flooding and compared corrosion inhibitors and corrosion-resistant alloys with an economic model [19]. Moreover, they also analyzed the failure of the acid gas wells and selected suitable materials for these gas wells [20]. Although many scholars have put forward different anti-corrosion measures for different environments [21–24], acid gas reinjection flooding still lacks corresponding anti-corrosion scheme. Due to the service, the environment is extremely severe, and only suitable materials should be selected to be tubing [25–28]. Oilfield companies try to use alloy steel as tubing to reduce the risk of corrosion, but it is difficult to effectively control the production cost of tubing [29,30]. What is more, it is necessary to reduce the cost of casing materials through wellbore design. It is the most important route to reduce the cost of tubing through reasonable wellbore design. Unfortunately, there is still a lack of relevant literature to integrate wellbore design methods and material selection.

In order to ensure the safe implementation of acid reinjection flooding, it is necessary to select appropriate materials for wellhead, wellbore, and downhole tools, combined with wellbore design, to form the best material selection scheme. In this paper, firstly, a trace of the literature is investigated to determine the steel used for wellhead, wellbore, and packer in an acid environment. Then, a weight-loss test is used to simulate the acid reinjection flooding environment for evaluating the steel corrosion, and surface characterization techniques are carried out to understand the corrosion risk in the production process. According to the residual tensile strength and internal pressure strength, the corrosion life of tubing is predicted, combing the economic model to calculate the cost of tubing. A new design scheme of a downhole string is proposed to control the cost of oil well string. Finally, the materials suitable for the wellhead, wellbore, and packer of reinjection acid gas drive wells are selected, and a new method of combining wellbore and materials is designed, which is beneficial to reducing costs.

## 2. Material Selection Basis

### 2.1. Basic Data Analysis of Acid Gas Reinjection Well

The good depth of the acid gas injection well located in Tahe oilfield is more than 5000 m. The formation pressure of the acid gas injection well is 55 MPa, and the wellhead pressure is 19–25 MPa, in which the content of $H_2S$ is 45% and the content of $CO_2$ is 55%. The material selection of key parts of an acid gas injection well, such as a wellhead, tubing, and downhole tools, needs to be studied systematically due to the corrosivity of acid gas.

### 2.2. Preliminary Selection of Materials

2.2.1. Material Selection Process for Wellhead Equipment of Gas Well

The selection steps of wellhead equipment materials are as follows:

(1) The data of composition and production of produced fluid, wellbore pressure and temperature are collected.
(2) According to the gas content and wellbore pressure data of $CO_2$ and $H_2S$ in the produced fluid, the gas partial pressure of $CO_2$ and $H_2S$ is calculated.

(3)  Based on the $CO_2$ and $H_2S$ partial pressure data, the application environment of wellhead equipment and a gas tree are classified to determine the corrosion degree of the application environment.

(4)  According to the classification of the application environment, the suitable material for wellhead equipment is selected to ensure that the wellhead equipment is safe and reliable.

The relevant provisions of ISO 10423 standard on the selection of wellhead materials are exhibited in Figure 1 [31]. According to the content of $CO_2$ and $H_2S$ at the wellhead, FF or HH gas tree shall be used for the wellhead, and stainless steel or corrosion-resistant alloy shall be used between the valve plate, valve seat, and valve body. Hence, 35CrMo steel, 825, 718, and 625 alloy are preliminarily selected for further verification in the laboratory.

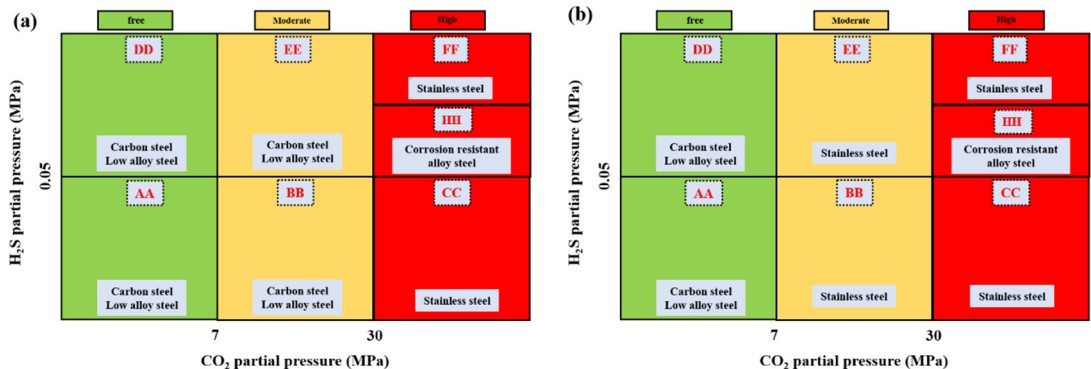

**Figure 1.** Application environment classification and material selection of wellhead. (**a**): body, cover, port, and outlet connection; (**b**): Pressure control parts.

### 2.2.2. Material Selection Process for Wellhead Equipment of Gas Well

Figure 2 displays the material selection chart of tubing in Q/SH 0015 standard [32]. According to the Q/SH 0015 standard, nickel base alloy steel ought to be selected for tubing. However, if the tubing is made of nickel base alloy, the cost of the wellbore will increase sharply. Therefore, we hope to use nickel base alloy (G3) and sulfur-resistant steel (T95) in combination through well-completion design to reduce the cost of well-completion. However, it is necessary to further evaluate G3 alloy and T95 steel in the proposed gas injection environment.

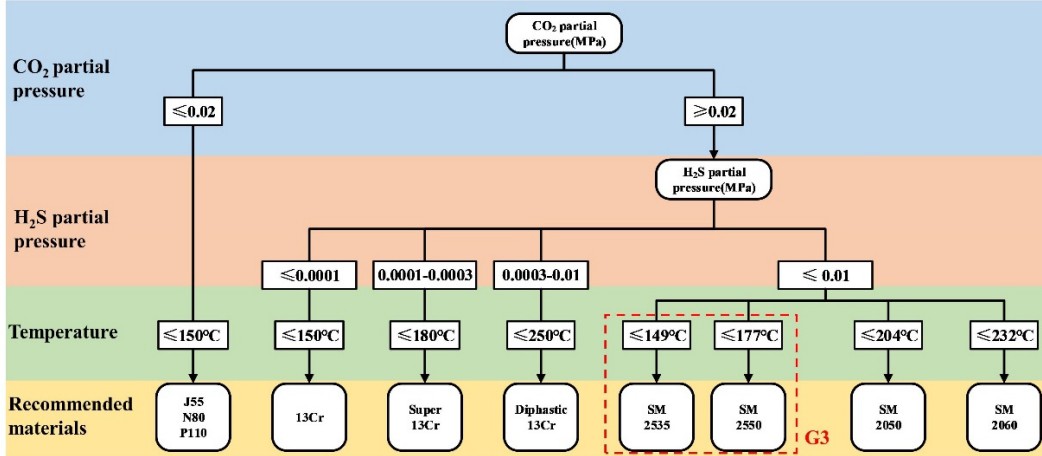

**Figure 2.** Material selection chart of tubing in Q/SH 0015 standard. Note: J55, N80, and 13Cr are widely used in the oil industry. SM2535 represents UNS N08535, SM2550 represents UNS N07750, and G3 represents UNS N06985. All materials adopt Q/SH 0015-2006 standard and API Specification 5CT or ISO 11960 2014 standard.

### 2.2.3. Material Selection Process for Packer of Gas Well

The low alloy steel used for the packer, such as 35CrMo and 42CrMo, is selected for non-corrosive wells or wells with a trace of $H_2S$. Stainless steel (such as 9Cr, 13Cr, super 13Cr) and corrosion-resistant alloy as packers are selected for wells with little $H_2S$ and $CO_2$. The nickel-based alloy to be as a packer, such as 825, 925, and 718, is selected for wells with severe $H_2S$ and $CO_2$. In view of the service environment of the acid gas reinjection flooding packer, 718 Alloy is selected for further testing, and 35CrMo is also selected for comparative study.

### 2.3. Material Selection Experiment

### 2.3.1. Determination of Service Environment of Wellhead, Wellbore, and Packer

To analyze the operating parameters of acid gas injection, the experimental parameters of the wellhead, wellbore, and packer steels are determined, as shown in Table 1. A sketch of the wellhead steel, wellbore steel, and packer steel structure is shown in Figure 3.

**Table 1.** Test schemes.

|  | Total Pressure | Phase | Temperature | $CO_2$ Content | $H_2S$ Content | Materials |
|---|---|---|---|---|---|---|
| wellhead | 24 MPa | liquid | 30 | 55% | 45% | 35CrMo, 825, 718, 625 |
|  |  | gas |  |  |  |  |
| wellbore | 55 MPa | liquid | 60, 90,120, 150 | 55% | 45% | T95, G3 |
|  |  | gas |  |  |  |  |
| packer | 55 MPa | liquid | 90 | 55% | 45% | 35CrMo, 718 |
|  |  | gas |  |  |  |  |

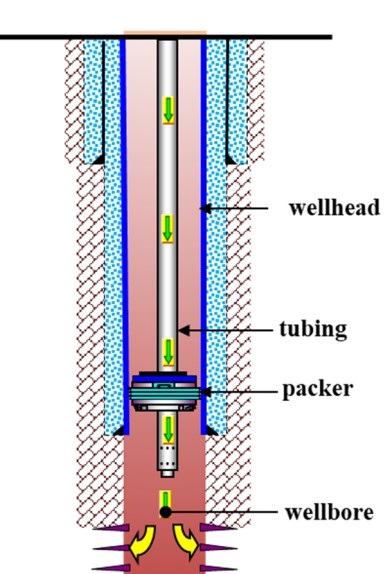

**Figure 3.** Wellbore, wellbore, and packer structure diagram.

### 2.3.2. Materials and Solution

Test samples, including carbon steel (35CrMo, T95) and nickel base alloy (G3, 825,718, and 625 steel) cut from tubing or downhole tools are provided by Tahe Oilfield, and their chemical composition is listed in Table 2. The samples are flat with a size of 30 mm × 15 mm × 3 mm according to ASTM standards, and the surface of the sample is polished by using SiC sandpaper (200#~1200#) to eliminate machining scratches. The samples are degreased with petroleum ether, washed with alcohol, and dried with cold air.



**Table 2.** Chemical composition of N80 and 3Cr steels (wt.%).

|  | Steel | C | Si | Mn | P | S | Cr | Mo | Ni | Ti | V | Al | Fe |
|---|---|---|---|---|---|---|---|---|---|---|---|---|---|
| Wellbore or Packer | 35CrMo | 0.35 | 0.22 | 0.50 | 0.02 | 0.03 | 0.85 | 0.2 | 0.2 | - | - | - | Bal. |
|  | 825 | 0.02 | 0.10 | 0.51 | - | - | 21.75 | 3.25 | 41.94 | 2.07 | - | 0.18 | Bal. |
|  | 718 625 | 0.033 | 0.14 | 0.065 | 0.0024 | 0.0006 | 18.96 | 3.28 | Bal. | - | - | - | 18.67 |
| Tubing | T95 | 0.30 | 0.22 | 0.52 | 0.01 | 0.01 | 0.99 | 0.17 | 0.01 | 0.02 | 0.007 | 0.22 | Bal. |
|  | G3 | 0.02 | 1 | - | 0.04 | 0.03 | 22 | 7 | Bal. | - | - | 0.20 | 20 |

### 2.3.3. Weight-Loss Test

A self-designed high temperature and high pressure (HTHP) autoclave made of C276 alloy is used for simulated corrosion testing (Figure 4).

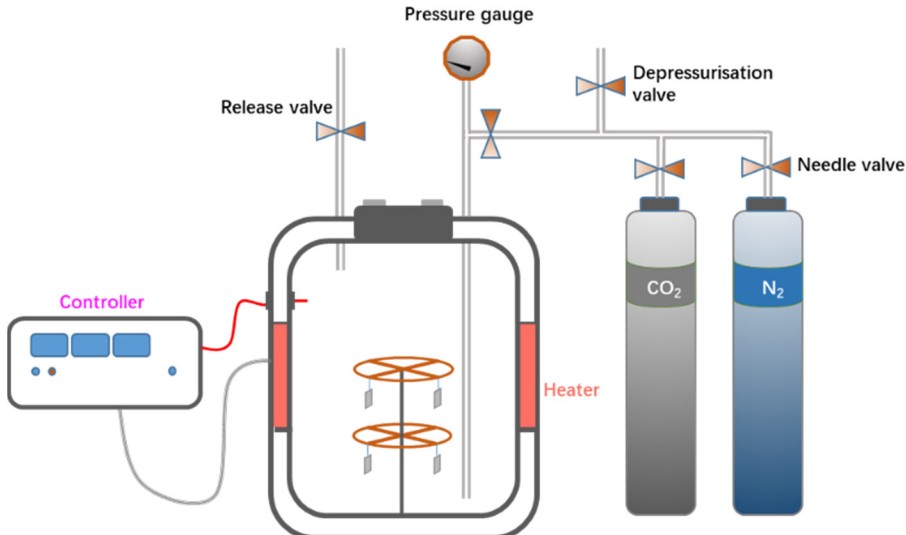

**Figure 4.** HTHP autoclave.

Before the experiment, the sample is firstly placed on the HTHP autoclave test piece fixture. Then, an appropriate amount of corrosive solution was added to the autoclave so that some samples were in the gas phase environment, and the rest of the samples were completely immersed in the solution (Figure 4). Then pure $N_2$ was introduced into the sealed HTHP autoclave to deoxygenate for 3 h. After turning on the reactor, set the temperature to the test temperature, pass $CO_2$ and $H_2S$ gas to the experimental pressure, and finally turn on the power supply of the reactor to start the experiment. The experimental parameters are shown in Table 1.

After the experiment, the samples were taken out from the HTHP autoclave, and a part of them was washed with deionized water and then dried in cold air. 100 mL hydrochloric acid (1.19 g/cm$^3$), 900 mL distilled water, and 10 g hexamethylenetetramine were prepared into a film removal solution (soak the test piece in the film removal solution and let it stand for a period of time. Wipe the test piece gently with a dust-free cloth to make the corrosion products on the surface of the test piece fall off), and the above samples were washed with the film removal solution to remove corrosion scales on the surface of the samples. The experimental samples were then washed and dehydrated with distilled water and alcohol, in turn and finally dried in cold air. The rest of the samples were used to observe the corrosion morphology and analyze the composition of corrosion scales.

Before and after the experiment, the mass of the samples was weighed with an electronic balance with an accuracy of 0.1 mg, and the corrosion rate of the samples was calculated according to Equation (1).

$$C_R = 87,600 \frac{\Delta m}{\rho A \Delta t} \tag{1}$$

where; $C_R$ is corrosion rate per year (mm/y), $\Delta m$ is weight loss (g), $\rho$ is material density (g/cm$^3$), $A$ is the total exposure surface area (cm$^2$), and $\Delta t$ is the total exposure time (h).

A Three-dimensional microscope (3D optical microscope, Bruker Contour GT-K) is used to observe the surface morphology of the samples after removing the corrosion products and test the local corrosion depth. According to the test results of local corrosion depth, the local corrosion rate is calculated by Equation (2).

$$R_L = \frac{0.365 \times h}{t} \tag{2}$$

where: $R_L$ is the local corrosion rate value (mm/a); $h$ is the maximum pitting depth (μm); $t$ is corrosion time (d).

### 2.3.4. Surface Characterization

A Japanese scanning electron microscope (Jeol, SEM JSM-6510A, Japan) was used to observe the surface and cross-sectional morphology of the samples, and the element distribution of the corrosion products was analyzed by energy dispersive spectrometer (EDS) with an accelerating voltage of 20 KV. The specific components of the corrosion products were analyzed by X-ray diffraction (CuK$\alpha$, $\lambda$ = 0.154 mm, Rigaku XRD, Model D/Max-B, Japan). 3D surface morphology after the removal of corrosion scales is observed by using a confocal laser microscope (Olympus OLS400, Japan).

### 2.3.5. Finite Service Life Calculation of Corrosion-Resistant Alloys and Carbon Steel Tubing

According to the weight-loss corrosion rate and pitting corrosion rate, the remaining corrosion life calculation model of tubing is established [33–35]. The core idea of the theory is that when the remaining wall thickness reaches its minimum allowable wall thickness, the service time of the tubing is its finite life. The corrosion type can be determined according to the surface morphology after tubing corrosion. The weight-loss corrosion rate and pitting corrosion are used to calculate the residual internal pressure and residual tensile strength. The residual corrosion life of tubing made of T95 and G3 steels is predicted by taking the threshold of tensile safety factor 1.2 and internal pressure safety factor 1.15, respectively.

The depth of the reinjection well is 4700 m, and the dimension of the wellbore tubing string is shown in Table 3.

**Table 3.** Tubing string size of acid injection gas well.

| Well Section | Tubing Size |
|:---:|:---:|
| 0 m~2000 m | φ 88.9 mm × 7.8 mm |
| 2000 m~4700 m | φ 73 mm × 5.5 mm |

(1)    Life assessment considering the tensile safety factor

Based on the size of the tubing, the residual tensile strength of the tubing is calculated. Then, the axial tension is calculated according to the API 5C3 standard, as shown in Equation (3).

$$T = \sigma S \tag{3}$$

The inner diameter of the tubing r becomes $r = r_0 + vt$ when the tubing has served for t time, and the axial tension becomes as follows:

$$T = \pi\sigma\left[R^2 - (r_0 + vt)^2\right]/4 \tag{4}$$

The condition for the safe operation of the oil pipe is that the axial stress does not exceed the yield strength of the materials, as shown in Equation (5).

$$\sigma_y > \sigma = \frac{T}{\pi\left[R^2 - (r_0 + vt)^2\right]/4} \tag{5}$$

The residual tensile strength of the tubing is as follows:

$$T_c = \pi\sigma_y\left[R^2 - (r_0 + vt)^2\right]/4 \tag{6}$$

where: $T$ is the axial tension of the tubing (KN); $v$ is the corrosion rate (mm/y); $\sigma$ is the axial stress (MPa); $S$ is the cross-sectional area of the tubing (cm$^2$); $t$ is the service time (h); $R$ is the external diameter of original tubing (mm); $r_0$ is the inner diameter of original tubing (mm); $\sigma_y$ is the yield strength (MPa); $T_c$ is the residual axial tension of tubing (KN).

(2)　Life assessment considering the internal pressure safety factor

According to the API 5C3 standard, the circumferential stress of tubing is calculated according to Equation (7).

$$\sigma = \frac{P_i R}{2\delta} \tag{7}$$

The wall thickness of the tubing becomes $\delta_0$ ($\delta_0 = \delta - vt$) after using t time. The circumferential stress of the tubing is calculated, as exhibited in Equation (8).

$$\sigma = \frac{P_i R}{2(\delta - vt)} \tag{8}$$

The condition for the safe operation of the oil pipe is that the circumferential stress does not exceed the yield strength of the materials, as displayed in Equation (9).

$$P_{bo} = \frac{2\sigma_y(\delta - vt)}{R} \tag{9}$$

where: $\delta$ represents the original wall thickness of the tubing, mm; $\delta_0$ represents the wall thickness of tubing after service time $t$, mm; $\sigma_y$ meaning the yield strength, MPa; $t$ is the service time, year; $P_i$ is the internal pressure on tubing, MPa; $R$ is the external diameter of original tubing, mm; $v$ is the corrosion rate, mm/y; $\sigma$ is the circumferential stress of tubing after service time $t$, MPa; $P_{bo}$ is the residual internal pressure strength of tubing, MPa.

## 3. Results

### 3.1. Corrosion Rate

The corrosion rates of carbon steel (35CrMo steel) and corrosion-resistant alloy steel (825, 718, 625 steel) used for wellhead in the gas and liquid phase are exhibited in Figure 5. Obviously, the corrosion rates of corrosion-resistant steel are much less than those of carbon steel. Moreover, the corrosion rate of carbon steel (35CrMo) corroded in a liquid environment is more serious, even exceeding the corrosion control value of the oilfield (0.076 mm/y).

Figure 6 illustrates the corrosion rates of T95 and G3 steel used for tubing in the gas and liquid phase. As the temperature increase, the corrosion rate of T95 steel corroded in the gas phase continue to rise, while that of G3 steel remains unchanged. Remarkably, once the temperature exceeds 120 °C, the corrosion rate of T95 steel increases slowly. The corrosion rates of T95 steel corroded in the liquid phase first rise and then decreases along

with temperature increases, while the corrosion rate of G3 is still unchanged, as given in Figure 3.

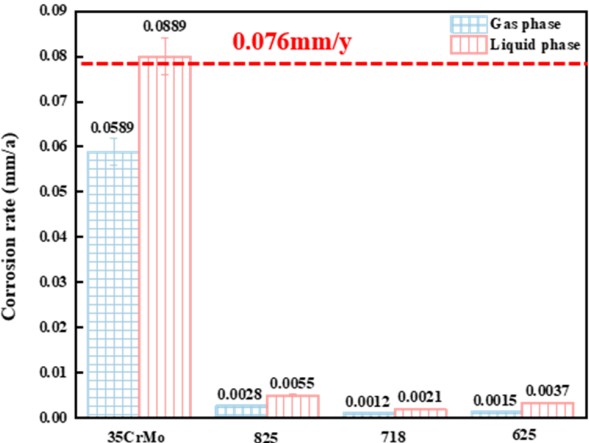

**Figure 5.** The corrosion rate of carbon steel (35CrMo steel) and the corrosion-resistant alloy steel (825, 718, 625 steel) used for wellhead in the gas and liquid phase.

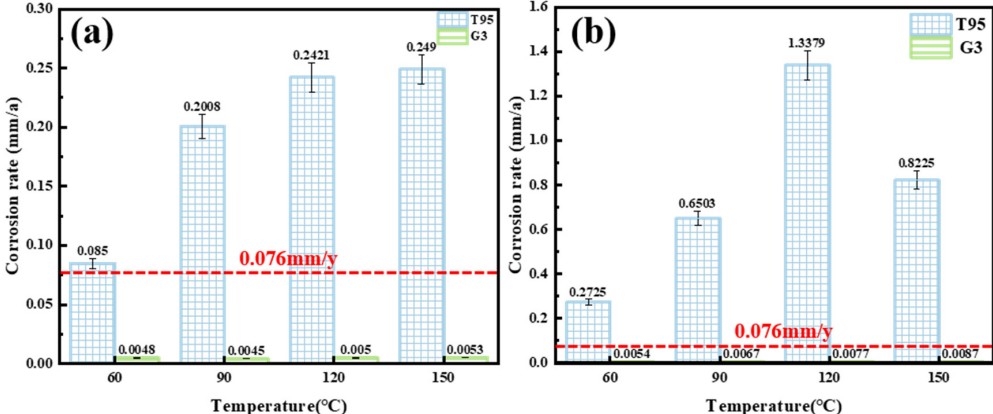

**Figure 6.** The corrosion rate of tubing in the gas and liquid phase ((**a**,**b**): gas and liquid phase).

Figure 7 shows carbon steel's corrosion rate and 718 alloys used for the packer in the gas and liquid phase. Obviously, the corrosion rates of 718 alloy steel are much less than those of carbon steel. Moreover, the corrosion rate of carbon steel (35CrMo) corroded in a liquid environment is more serious, even exceeding the control value of the oilfield.

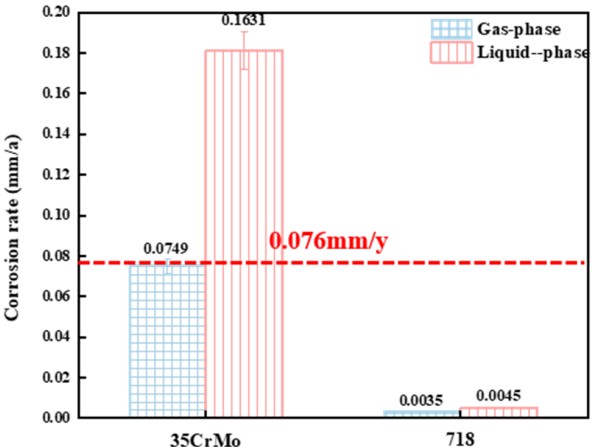

**Figure 7.** The corrosion rate of carbon steel and 718 alloys used for the packer in the gas and liquid phase.

### 3.2. Characteristics of Corrosion Scales

3.2.1. Surface Morphology

The surface of corrosion-resistant alloy steel is still bright after corrosion, explaining that steel has good corrosion resistance in an acid environment. Hence, only the surface and micro-morphology of corrosion products formed after corrosion are observed.

The macro-morphology and micro-morphology of 35CrMo steel used for the wellhead are displayed in Figure 8. The surface of the samples in the liquid phase is completely covered with black corrosion products (Figure 8a), while that of samples in the gas phase is only partially covered, and other regions show metallic luster (Figure 8c). Significantly, the coniferous and lamellar corrosion products are observed in the liquid phase (Figure 8b), while the corrosion products in the gas phase environment are composed of hexagonal crystals (Figure 8d). Moreover, a tremendous of pores in the corrosion products are also observed, leading to the corrosive solution further corroding the samples.

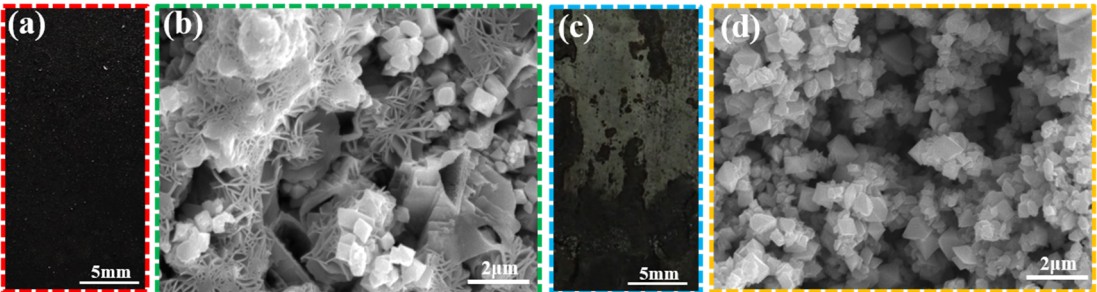

**Figure 8.** Macro-morphology and micro-morphology of 35CrMo steel used for wellhead. ((**a**,**b**): liquid phase; (**c**,**d**): gas phase).

The macro-morphology and micro-morphology of T95 steel used for wellbore at various temperatures in the gas phase are exhibited in Figure 9. From the results of macro-morphology, the surface of steel appears black in some areas, and even some areas turn blue. Micro-morphology results reveal that the corrosion products on the surface of steel are similar. In the wake of rise in temperature, the corrosion product films become dense, and the pores in the film reduce.

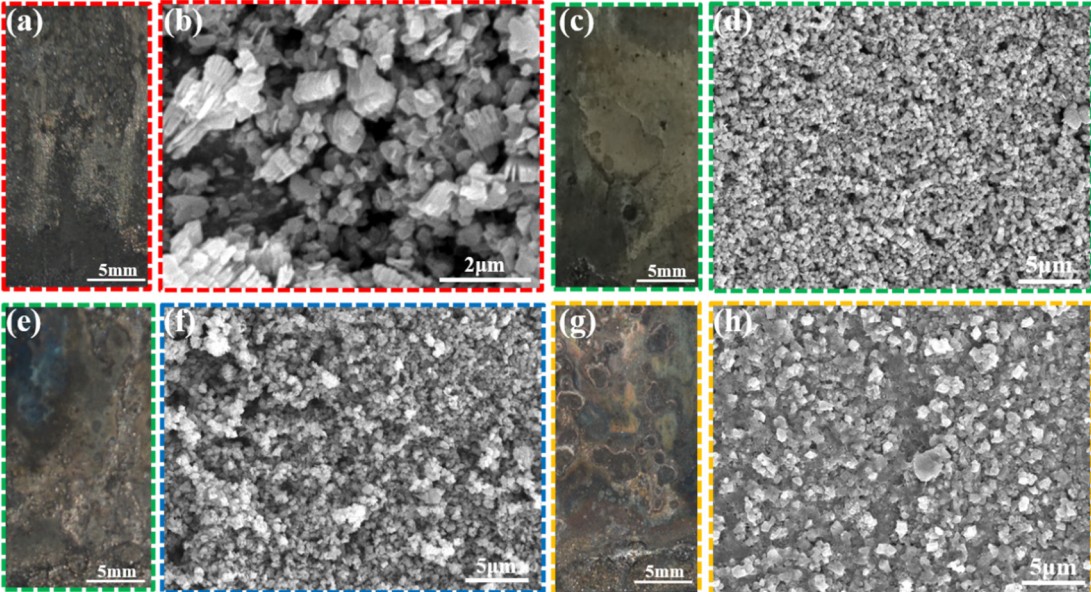

**Figure 9.** Macro-morphology and micro-morphology of T95 steel used for wellbore at various temperature in the gas phase ((**a**,**b**): 60 °C; (**c**,**d**): 90 °C; (**e**,**f**): 120 °C; (**g**,**h**): 150 °C).

The macro-morphology and micro-morphology of T95 steel used for wellbore at various temperatures in the liquid phase are exhibited in Figure 10. Interestingly, the color of the surface of samples varies greatly. Among them, the surface of the steel is blackest at 90 °C (Figure 10b) and becomes lighter at 120 °C and 150 °C (Figure 10c,d). One of the details is easy to observe is that a grainy feeling on the surface of samples, especially at 120 °C and 150 °C (Figure 10c,d). The cube and rod products are deposited on the surface of the samples. Meantime, the tiny products also surrounded the cube product at 60 °C (Figure 10a). The cubic and long rod-shaped crystals are covered on the surface of T95 steel in the liquid phase. Interestingly, many small particles also adhere to the cubic crystals.

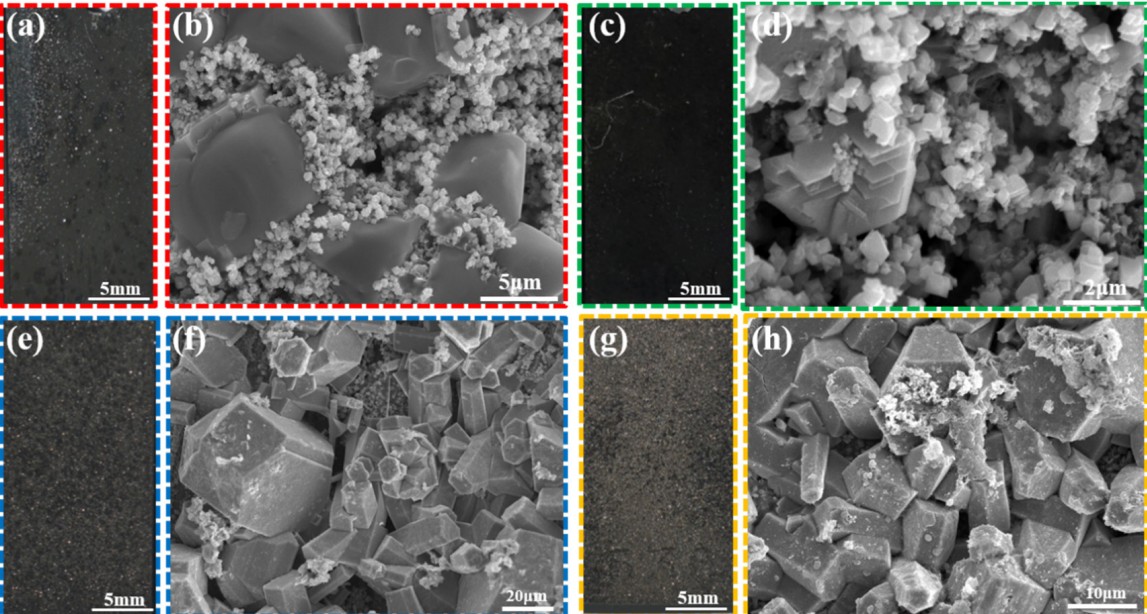

**Figure 10.** Macro-morphology and micro-morphology of T95 steel used for wellbore at various temperatures in the liquid phase ((**a**,**b**): 60 °C; (**c**,**d**): 90 °C; (**e**,**f**): 120 °C; (**g**,**h**): 150 °C).

The macro-morphology and micro-morphology of 35CrMo steel used for the packer at various temperatures in the gas and liquid phase are exhibited in Figure 11. Metallic luster can be observed on the surface of the steel in the gas phase, while black corrosion products are covered on the surface of the steel in the liquid phase. The corrosion product particles formed in the gas phase are quite small, and a few holes can be observed between the particles. Cubic corrosion product particles are stacked on the surface of the steel in the liquid phase. Moreover, there are many holes around the corrosion products.

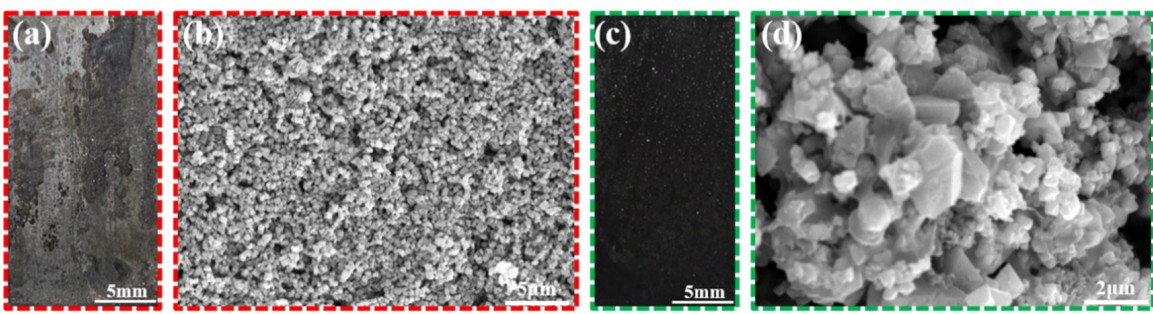

**Figure 11.** Macro-morphology and micro-morphology of T95 steel used for packer in gas (**a**,**b**) and liquid (**c**,**d**) phase.

### 3.2.2. 3D Surface Morphology and Pitting Rate

Surface and 3D morphology (3D optical microscope, Bruker ContourGT−K, MeimingIC, Germany) of T95 steel samples in the liquid phase after washing and removing the corrosion products are shown in Figure 12. The local corrosion rate of T95 steel in the liquid phase is shown in Figure 13. The surface of samples in the liquid phase is densely covered with pitting, especially at 90 °C. As the temperature increases, the number and depth of pitting decline, implying that the low temperature is more beneficial to the formation of pitting.

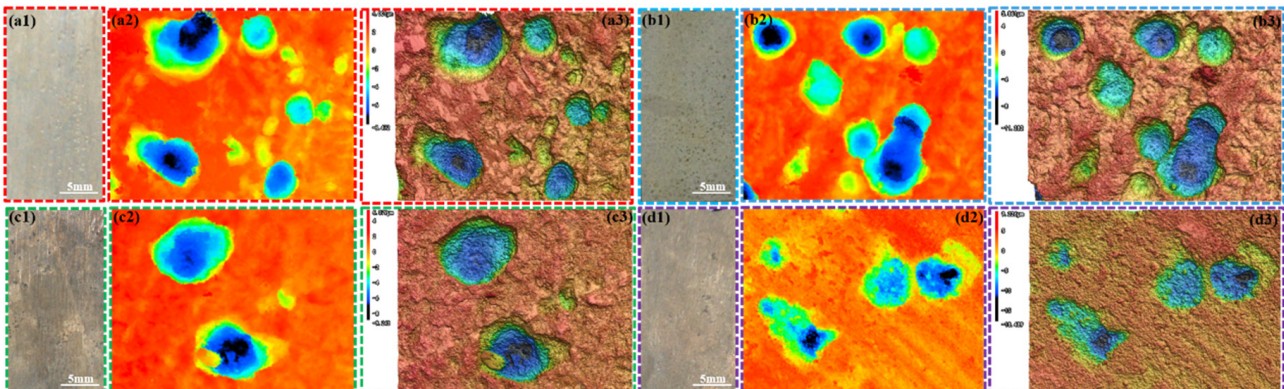

**Figure 12.** Surface and 3D morphology of T95 steel corroded in the liquid phase after cleaning the corrosion products ((**a–d**) is 60 °C, 90 °C, 120 °C, 150 °C).

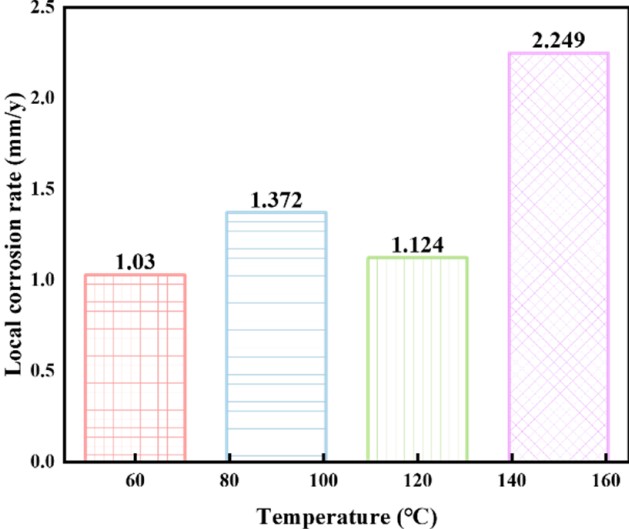

**Figure 13.** The local corrosion rate of T95 steel corroded in the liquid phase.

### 3.2.3. Element Distribution

The pitting corrosion is an important factor in causing tubing and casing failure. Therefore, to further explore the pitting corrosion in a liquid environment, the element distribution in the pitting is analyzed by element distribution (EDS), as shown in Figure 14. The pitting is rich in the O element, while the S element is mainly concentrated in the outer layer of the O corrosion product film, indicating the pitting corrosion is caused by $CO_2$ corrosion.

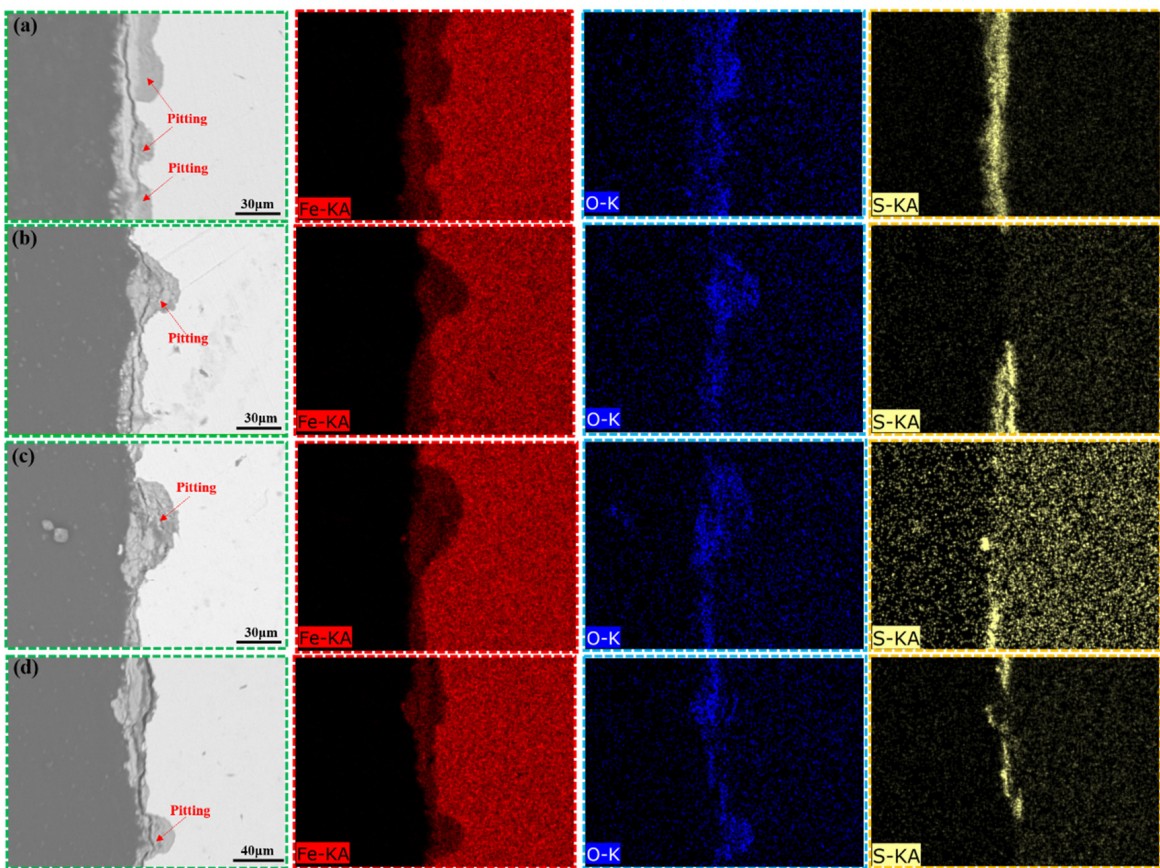

**Figure 14.** Element distribution in the pitting of T95 steel corroded in the liquid phase ((**a–d**) is 60 °C, 90 °C,120 °C,150 °C).

### 3.2.4. Phase Composition of Corrosion Products

An X-ray diffractometer was used to analyze the phase composition of the corrosion products of the corroded samples to obtain the XRD pattern. Jade software was used to compare and analyze the diffraction peak pattern obtained with the reference material card so as to judge the phase composition of the corrosion products of the samples. The phase composition of corrosion scales of T95 steel in the liquid and gas phases are shown in Figure 15. XRD results demonstrated that the corrosion products of carbon steel are mackinawite and iron sulfide (FeS), and Ferrous carbonate ($FeCO_3$).

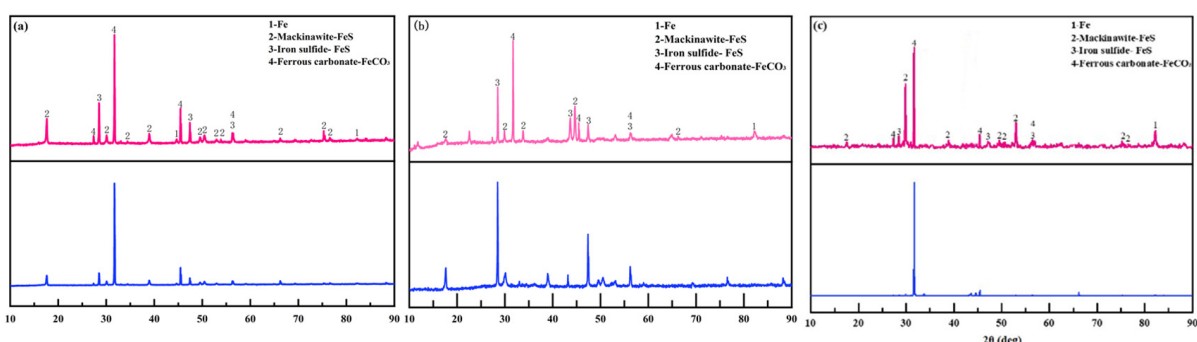

**Figure 15.** Phase composition of corrosion scales of T95 steel in the liquid and gas phase ((**a–c**) is 90 °C,120 °C,150 °C).

### 3.3. Safe Service Life of Carbon Steel and Nickel Base Alloy Steel

3.3.1. Service Life of Tubing Considering Uniform Corrosion

Based on the theoretical basis of the safe service life of casing, the remaining safe service life in different temperatures is predicted. Figures 16 and 17 described the residual collapse strength and residual tensile strength of T95 steel and G3 alloy in gas and liquid phases at various temperatures. The tensile safety factor and internal pressure safety factor of G3 alloy tubing are much greater than those of T95 steel. Therefore, the safety and service life of G3 alloy tubing meets the requirements. The safe service life of T95 steel is determined according to the critical internal pressure and tensile safety factor, as plotted in Figure 17. T95 steel tubing considering the tensile strength, can safely serve for 20 years in the gas phase while only for 3.7 years in the liquid phase. T95 steel tubing considering internal pressure strength can safely serve for 8.7 years in the gas phase while only for 1.6 years in the liquid phase.

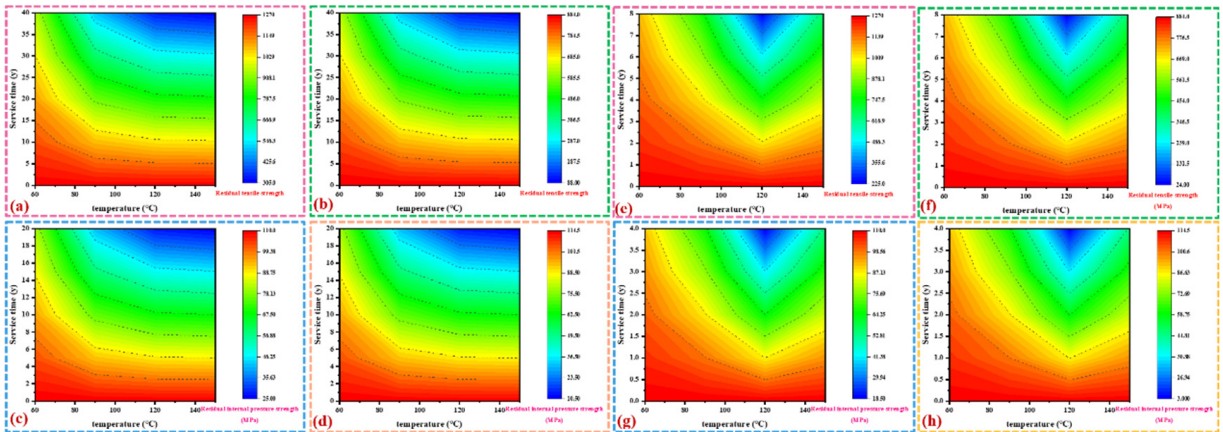

**Figure 16.** Residual tensile strength of T95 (**a**,**c**,**e**,**g**) and G3 (**b**,**d**,**f**,**h**) steels in gas (**a**,**b**,**e**,**f**) and liquid (**c**,**d**,**g**,**h**) phase ((**a**–**d**): 0–2000 m; (**e**–**h**): 2000–4700 m).

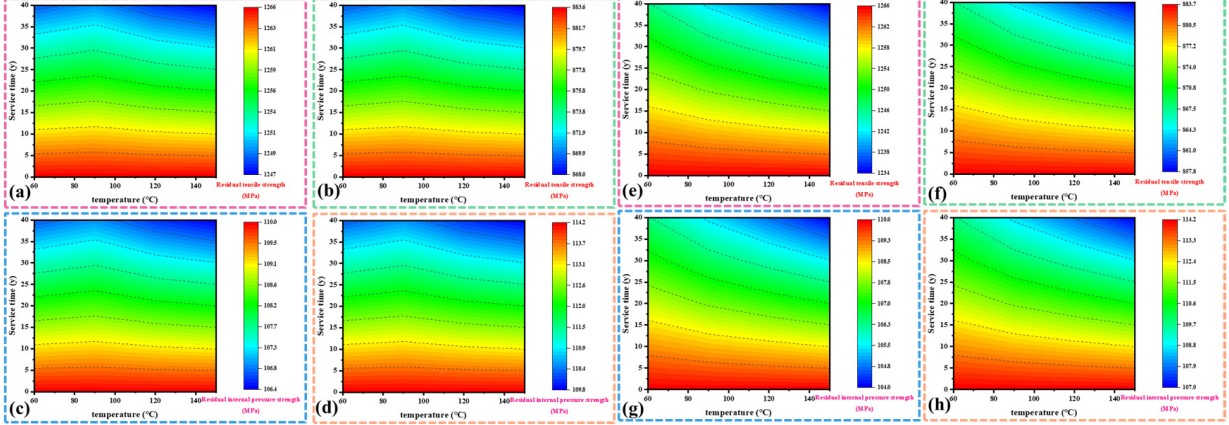

**Figure 17.** Residual collapse strength of G3 (**a**,**c**,**e**,**g**) and G3 (**b**,**d**,**f**,**h**) steels in gas (**a**,**b**,**e**,**f**) and liquid (**c**,**d**,**g**,**h**) phase ((**a**–**d**): 0–2000 m; (**e**–**h**): 2000–4700 m).

The safe service life of T95 steel is determined according to the critical internal pressure and critical tensile safety factor, as plotted in Figure 18. The safe service life of T95 steel in a liquid environment is less than that in a gas environment. The safe service life of gas phase must be considered to ensure the safe use of tubing. Therefore, the safe service life of T95 steel as the tubing is 1.63 years.

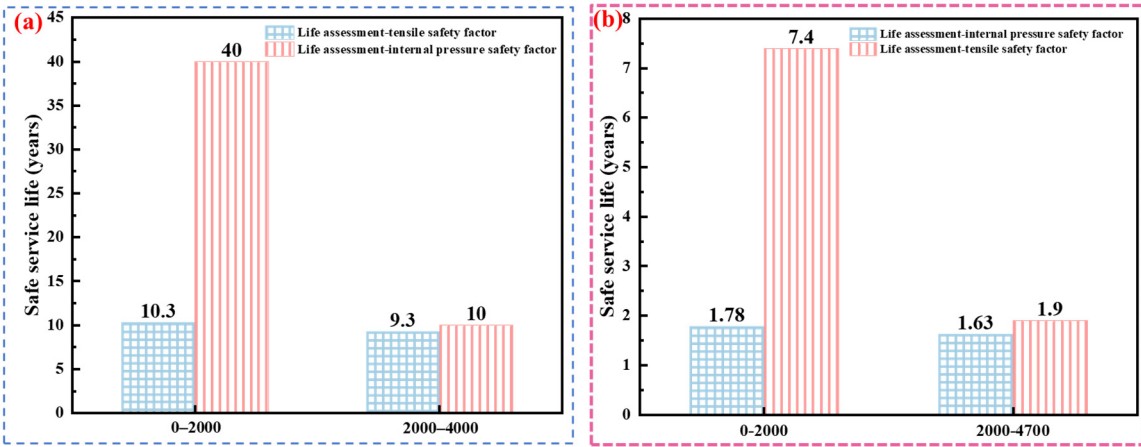

**Figure 18.** The safe service life of T95 steel in the gas (**a**) and liquid (**b**) phases.

### 3.3.2. Service Life of Tubing Considering Pitting Corrosion

Due to the serious pitting corrosion of T95 steel in a liquid environment, the safe service life of tubing can be predicted according to the local corrosion rate. Figure 19 describes the residual tensile strength and residual internal pressure strength of T95 steel considering the pitting corrosion in gas and liquid phases at various temperatures, and the safe service life of T95 steel is determined according to the critical internal pressure and critical tensile safety factor, as shown in Figure 20. For the sake of safety, the safe service life of T95 steel is only one year.

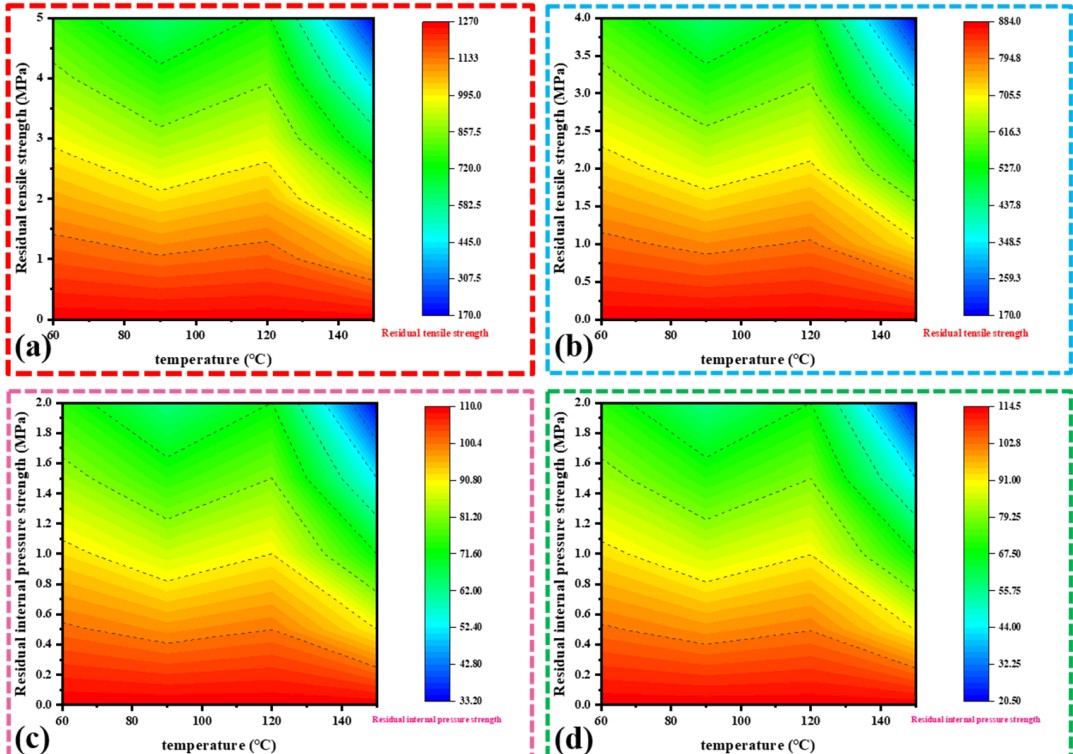

**Figure 19.** Residual tensile strength of T95 steels considering pitting corrosion: 0–2000 m; 2000–4700 m ((**a,c**): Residual tensile strength and Residual internal pressure strength in gas phase; (**b,d**): Residual tensile strength and Residual internal pressure strength in liquid phase).

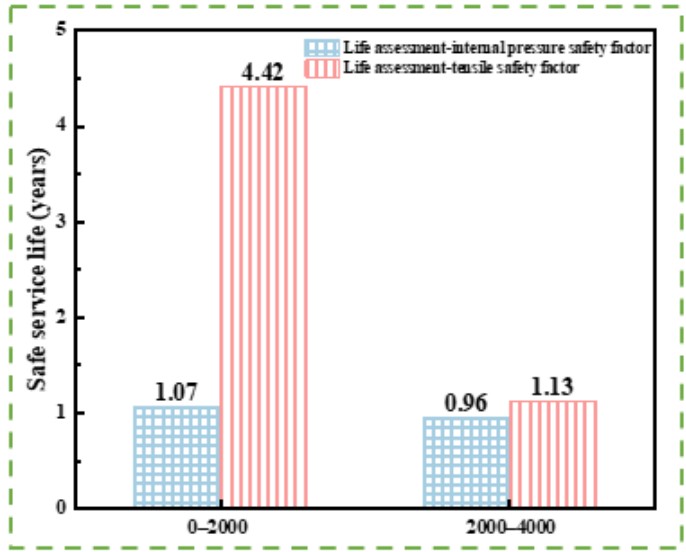

**Figure 20.** The safe service life of T95 steel in the liquid phase considering pitting corrosion.

## 4. Discussion

### 4.1. Pitting Corrosion of T95 Steel in Liquid Phase

In the liquid phase, $CO_2$ and $H_2S$ dissolve in the solution to produce the corrosive ions, including $HS^-$, $S^{2-}$, $CO_3^{2-}$, and $HCO_3^-$ ions [36,37]. The cathodic and anodic reactions in the solution are as follows [38].

$$2H_2S + 2e^- \rightarrow 2HS^- + H_2 \tag{10}$$

$$2HS^- + 2e^- \rightarrow 2S^{2-} + H_2 \tag{11}$$

$$2H^+ + 2e^- \rightarrow H_2 \tag{12}$$

$$2H_2CO_3 + 2e^- \rightarrow 2HCO_3^- + H_2 \tag{13}$$

$$2HCO_3^- + 2e^- \rightarrow 2CO_3^{2-} + H_2 \tag{14}$$

$$Fe \rightarrow Fe^{2+} + 2e^- \tag{15}$$

Figure 21 reveals the schematic diagram of the mechanism of pitting corrosion of T95 steel in the liquid phase. The corrosion product films of carbon steel forming under a $CO_2$-$H_2S$ environment usually are composed of FeS and $FeCO_3$ [39]. FeS and $FeCO_3$ deposit on the surface of carbon steel to form the corrosion product films when the concentrations of $[Fe^{2+}] \times [S^{2-}]$ and $[Fe^{2+}] \times [CO_3^{2-}]$ respectively exceed the solubility of FeS and $FeCO_3$ (Equations (16) and (17)) [40,41]. However, since the supersaturation of FeS crystals is way below than $FeCO_3$ crystals (25 °C, $K_{spFes} = 6.3 \times 10^{-18}$, $K_{spFeCO_3} = 3.2 \times 10^{-11}$) [30], FeS crystals are more easily to deposit on most areas of the surface, while $FeCO_3$ is the mainly deposited on a few areas (Figure 21b). Because the FeS layer is extremely dense, it is difficult for chloride ions to penetrate through the FeS layer to erode the steel. However, because of the pores between $FeCO_3$ crystals, chloride ions can easily pass through the loose $FeCO_3$ films and continue to erode the steel, forming pitting corrosion [42] (Figure 21c).

$$Fe^{2+} + S^{2-} \rightarrow FeS \tag{16}$$

$$Fe^{2+} + CO_3^{2-} \rightarrow FeCO_3 \tag{17}$$

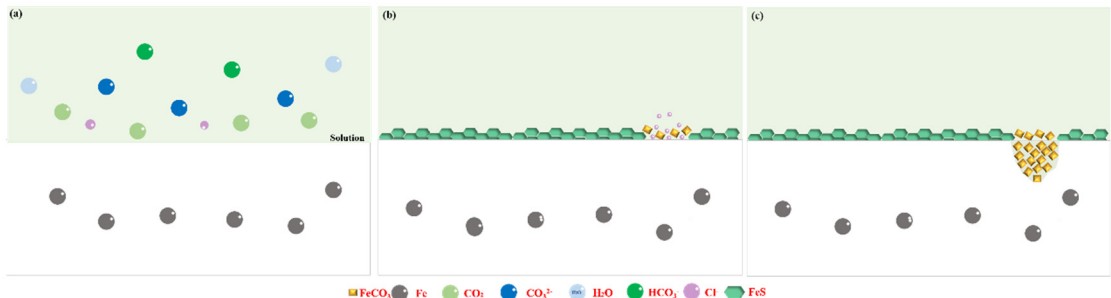

**Figure 21.** Schematic diagram of the mechanism of pitting corrosion of T95 steel in the liquid phase ((**a**): Interaction of ions with metal substrates; (**b**): Deposition of FeS and $FeCO_3$; (**c**): $Cl^-$ pass through the corrosion product film to form pitting corrosion).

*4.2. Economic Cost Calculation of Tubing Made of Carbon Steel and Alloy Steel*

Any anti-corrosion measures must consider economic factors in the anti-corrosion scheme of gas injection well in acid gas reinjection flooding. On the basis of ensuring the anti-corrosion effect of the gas injection well, the economic benefits must be maximized during the gas injection process.

The weight of the tubing is calculated as follows:

$$M = \pi \times h \times \rho \frac{(D^2 - d^2)}{4} \times 10^{-6} \tag{18}$$

where: $M$ is the weight of the tubing, ton; $D$ is the outside diameter of the tubing, mm; $d$ is the inside diameter of the tubing, mm; $h$ is the length of the tubing, m; $\rho$ is the density of steel, $g/m^3$.

The price of a single well tubing is as follows:

$$P_t = P \times M \tag{19}$$

where: $P_t$ is the price of a single well tubing, ten thousand yuan; $P$ is the price per ton of steel (Table 4), ten thousand yuan.

**Table 4.** Price of tubing with different materials for a single well.

| Materials | $P$ (Ton/Ten Thousand Yuan) | $M$ (Ton) | $P_t$ (Ten Thousand Yuan) |
|-----------|------------------------------|-----------|----------------------------|
| T95 | 1.127 | 56.06 | 63.16 |
| G3 | 0.78 | 59.04 | 1771.2 |

Note: The unit price is quoted by the steel mills.

Table 4 calculates the cost of tubing with different materials in a single well. As it shows, the cost of G3 alloy steel tubing is much higher than that of carbon steel tubing. However, the corrosion risk of G3 steel tubing is significantly reduced.

Assuming that the service life of the injection well has been in service for 10 years, 20 years, and 30 years respectively, the total cost of using tubing of different materials in different service life is shown in Table 5. To ensure the safety of tubing, the safe service life of T95 steel considering the pitting corrosion rate in the liquid phase is adopted to calculate the tubing replacement frequency of 10 years, 20 years, and 30 years, respectively. In addition, the labor cost of replacing the tubing also must be considered ($C_L$). Table 5 reveals the total costs of changing tubing in different service periods. When T95 steel is used for 10 and 20 years, its cost is significantly lower than that of G3 steel. However, it is nearly close to G3 alloy when it has been used for more than 30 years.

**Table 5.** Total costs of changing tubing in different service periods (Ten thousand yuan).

| Materials | $P_t$ | $C_L$ | Tubing Replacement Frequency | | | Total Costs | | |
|---|---|---|---|---|---|---|---|---|
| | | | 10 Year | 20 Year | 30 Year | 10 Year | 20 Year | 30 Year |
| T95 | 63.16 | 10 | 10 | 20 | 30 | 631.6 | 1263.2 | 1894.8 |
| G3 | 1771.2 | 10 | 1 | 2 | 4 | 1771.2 | 1771.2 | 1771.2 |

*4.3. New Design Scheme of Downhole String*

According to the above analysis, the corrosion risk of T95 steel is severe, while the cost of G3 alloy is expensive. How to not only ensure the safe operation of tubing but also reduce the price of tubing has become the primary consideration. Therefore, it is necessary to design a new string structure for the downhole packer, as shown in Figure 22, which can not only ensure the safe operation of tubing but also reduce the price of tubing. The string structure of the downhole packer mainly includes a casing, casing packer, liner hanger, and liner, and the tubing string structure mainly includes tubing, single flow valve, tubing, single flow valve, tubing packer, tubing nipple, and trumpet assembly, in which the single flow valve B is located below the tubing packer. The first acid gas barrier is composed of single flow valve B, tubing B, tubing packer, liner, and cement sheath, and the second acid gas barrier consists of tubing A, single flow valve A, casing, casing packer, and annulus protection fluid. Among them, G3 alloy steel tubing is used between the packer and the relief valve, T95 steel tubing is selected above the packer and below the safety valve, and the packer is set in the G3 steel tubing. Two barriers provide a guarantee for preventing acid gas from channeling in the annulus and tubing. The new downhole packer can effectively isolate the acid gas in the near ground layer, prevent the acid gas from upward gas channeling, and effectively ensure the safety of the acid gas reinjection well.

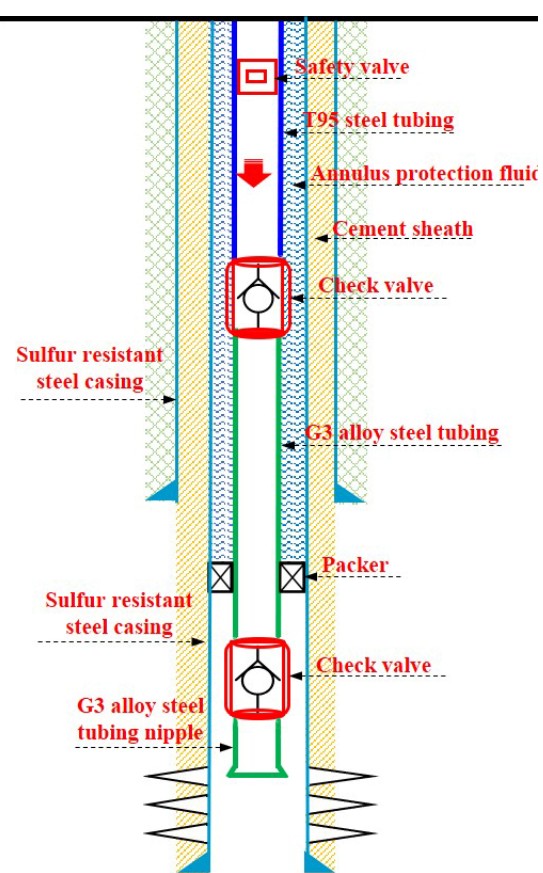

**Figure 22.** A new downhole packer for acid gas reinjection.

## 5. Conclusions

(1) 35CrMo steel is used for wellhead materials, 625 alloy steel is selected as the sealing surface, and 625 or 825 alloy steel is used for wellhead sealing material. 718 nickel base alloy is selected as packer steel.

(2) Its cost is significantly lower than that of G3 steel when T95 steel, as the tubing is serviced for 10 and 20 years. Its cost is nearly close to G3 alloy when it has been used for more than 30 years.

(3) A kind of downhole packer for acid gas reinjection is proposed, which has a double barrier to ensure the safety of wellbore gas injection. G3 alloy steel tubing is used between the packer and the relief valve, T95 steel tubing is selected above the packer and below the safety valve, and the packer is set in the G3 steel tubing.

(4) FeS is deposited on most of the steel surface, while $FeCO_3$ is only deposited on some parts of the steel surface. Chloride ion easily passes through the loose $FeCO_3$ films while is blocked by the dense FeS films.

**Author Contributions:** Conceptualization, Y.G. and Z.L.; methodology, Z.L.; validation, Y.F. and W.L.; formal analysis, Z.L.; resources, Y.G. and B.D.; data curation, W.Z. and Q.W.; writing—original draft preparation, Z.L.; writing—review and editing, Z.L.; All authors have read and agreed to the published version of the manuscript.

**Funding:** This research received no external funding.

**Conflicts of Interest:** The authors declare no conflict of interest.

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
