# Peer review of "How to Choose the Suitable Steel of Wellhead, Wellbore, and Downhole Tools for Acid Gas Reinjection Flooding"

_processes, doi:10.3390/pr10122685_

Round 1
Reviewer 1 Report
The presented study entitled How to choose the suitable steel of wellhead, wellbore and downhole tools for acid gas reinjection flooding is an interesting and logically organized study. I appreciate the fact that, according to the presented study, the authors implemented a simulation and based on it drew conclusions, which they discussed and searched for the cause in a relatively appropriate way. Comments and questions: 1. The abstract must be supplemented with basic quantifiable data from the results of the analysis. It is not correct to rely only on verbal conclusions. Fill in please. 2. The analysis could be supplemented with basic statistical indicators, or results about the value of selected types of materials need to be supplemented with quantifiable indicators that are compared by statistical tests, e.g. medium values. I am asking the authors to supplement the questionable points of the study for me.Author Response
Dear Editors and Reviewers:
Thank you for your letter and for the comments concerning our manuscript entitled “How to choose the suitable steel of wellhead, wellbore and downhole tools for acid gas reinjection flooding” (ID: processes-1951469). Thank you for giving me this chance to revise my paper. Those comments are all valuable and very helpful for revising and improving our paper, as well as the important guiding significance to our researches. We have studied comments carefully and have made correction which we hope meet with approval.
According to the comments of reviewers, revised portions are marked in red in the revised manuscript. The changes in the revised manuscript and the responses to the reviewer’s comments are as follows. Hope these will make it more acceptable for publication. Thank you and best regards.
Yours sincerely,
Name: Zhendong Liu, E-mail: liuzd0522@foxmail.com
Response to comments:
Q1: The abstract must be supplemented with basic quantifiable data from the results of the analysis. It is not correct to rely only on verbal conclusions. Fill in please.
Response: Thank you very much for the valuable advice. We have added basic quantifiable data to the abstract to support the conclusions in the abstract.
Q2: The analysis could be supplemented with basic statistical indicators, or results about the value of selected types of materials need to be supplemented with quantifiable indicators that are compared by statistical tests, e.g. medium values. I am asking the authors to supplement the questionable points of the study for me.
Response: Thank you for your comment. We have supplemented the abstract and text of the manuscript with references such as API Specification 5CT or ISO 11960 2014, GB/T 3077-2015 and ASTM B637-16. The experimental data of this paper aims to provide guidance for material selection at different positions of acid gas injection Wells by comparing corrosion rate analysis of different pipes under different experimental conditions. Material selection is recommended based on quantified corrosion rates and economic analysis results.

Reviewer 2 Report
The authors provide in their article a fascinating topic regarding the oilfield problems, they use an actual case study on the material selection and how it will affect the corrosion rates. the manuscript entitled How to choose the suitable steel of wellhead, wellbore, and downhole tools for acid gas reinjection flooding needs major revisions.
All my comments are provided in the attached file as notes.

Author Response
Dear Editors and Reviewers:
Thank you for your letter and for the comments concerning our manuscript entitled “How to choose the suitable steel of wellhead, wellbore and downhole tools for acid gas reinjection flooding” (ID: processes-1951469). Thank you for giving me this chance to revise my paper. Those comments are all valuable and very helpful for revising and improving our paper, as well as the important guiding significance to our researches. We have studied comments carefully and have made correction which we hope meet with approval.
According to the comments of reviewers, revised portions are marked in red in the revised manuscript. The changes in the revised manuscript and the responses to the reviewer’s comments are as follows. Hope these will make it more acceptable for publication. Thank you and best regards.
Yours sincerely,
Name: Zhendong Liu, E-mail: liuzd0522@foxmail.com
Response to comments:
Q1: Many abbreviations in the abstract are not well known for all readers, I recommend put these abbreviations in detail in abstract. What is the name of the 625 material? What is the standard used? G3, T95, ------What are these materials? In general, the abstract should be rewritten in good organized way.
Response: Thank you for your comment. We are very sorry that due to the lack of full names of the materials in the abstract, which make you cannot understand the meaning of these materials. Considering your multiple suggestions on the abstract, we put them together and responses. We have added a statement about the meaning of the material in abstract and rewritten the abstract in good organized way. These materials are commonly used in the petroleum industry, where 825 refers to UNS NO8825, 625 refers to UNS NO6625, G3 refers to hastelloy G3 (UNS N06985) and 718 refers to UNS 07718. API Specification 5CT or ISO 11960 2014, GB/T 3077-2015 and ASTM B637-16 are used for all metal materials. There are many similar expressions in the literature as follows:
[1] Al-Saadi, M., Hulme-Smith, C., Sandberg, F. et al. Hot Deformation Behaviour and Processing Map of Cast Alloy 825. J. of Materi Eng and Perform 30, 7770–7782 (2021).
[2] Hrishikesh Das, Mounarik Mondal, Sung-Tae Hong, et al. Texture and precipitation behavior of friction stir welded Inconel 825 alloy. Materials Today Communications, 2020, 25: 10195.
[3] Bai Q, Bian L, Zhao Q, et al. Effect of Grain Boundary Engineering on Intergranular Corrosion Resistance of Incoloy825 Alloy[J]. Corrosion & Protection, 2019.
The revised abstract is as follows:
The material selection of injection gas well in acid gas flooding is the bottleneck of the successful implementation of the technical scheme. Through standard and literature research, the materials of wellhead, wellbore and packer for reinjection well in acid gas flooding are preliminarily established, and then the suitable materials are further screened by using weight-loss and surface characterization method. Finally, a new type of packer is designed to optimize the wellbore material. The results shows that 35CrMo (CR=0.0589mm/y) steel is used for wellhead materials, 625 alloy steel is selected as the sealing surface and 625 or 825 alloy (with CR≤0.0055mm/y) steel is used for wellhead sealing material. The main material of packer is 718 Alloy (with CR≤0.0021mm/y). The cost of T95 steel within 20 years (1263 ten thousand yuan) of service is much smaller than that of G3 alloy (1771 ten thousand yuan), but after 30 years of service, its cost is close to that of G3 alloy. A kind of downhole packer for acid gas reinjection is designed. Among them, G3 alloy steel tubing is used between the packer and the relief valve, T95 steel tubing is selected above the packer and below the safety valve and the packer is setting in the G3 steel tubing. The serious pitting corrosion of T95 steel in the liquid phase environment is due to the uneven deposition of FeS and FeCO3 on the steel surface.
Q2: The cost of T95 steel within 20 years of service is much smaller than that of G3 alloy, but after 30 years of service, its cost is close to that of G3 alloy. How did you get this conclusion?
Response: Thank you for your comment. According to the calculation and analysis (line392-line400) in Table 5, when the service life exceeds 30 years, the total cost of T95 is 1894.8 ten thousand yuan, exceeding the total cost of G3 which is 1771.2 ten thousand yuan.
Q3: The last sentence needs to be revised (Line 80-82).
Response: Thank you for your comment. We revised this sentence and read it carefully to make sure it was correct. The revised sentence is as follows:
Finally, the materials suitable for the wellhead, wellbore, and packer of reinjection acid gas drive wells are selected, and a new method of combining wellbore and materials is designed, which is beneficial to reducing costs.
Q4: Why there are some paragraphs are highlighted in grey color?
Response: Thank you for your detailed comment. This may be due to the fact that the color annotations were not remove during the manuscript submission process. We have corrected this problem in the revised manuscript and resubmitted it.
Q5: All the mentioned alloys, their chemical compositions should be mentioned. Some language errors should be corrected.
Response: Thank you for your comment. All the mentioned chemical compositions of metal materials are presented in Table 2 in the manuscript for the reader's reading.
Q6: You should put the reference number here also (Line 109-110). I wanted to understand this chart, but I couldn't because using symbols not well defined in any place in the manuscript (Fig. 2).
Response: Thank you for your comment. We have added the reference number to the title of Figure 1. We changed the presentation of Fig. 2 to make it easier for the reader to read. The revised Fig.2 is shown below, and we have replaced the revised figure into the manuscript. In addition, due to our negligence in showing here the meaning of the symbol of the reform to the material, we have also added a note in the manuscript. The supplemented manuscript is as follows:
Fig.2 Material selection chart of tubing in Q/SH 0015 standard
Note: J55, N80, and 13Cr are widely used in the oil industry. SM2535 represents UNS N08535, SM2550 represents UNS N07750, and G3 represents UNS N06985. All materials adopt Q/SH 0015-2006 standard and API Specification 5CT or ISO 11960 2014 standard.
Q7: Please draw a sketch to illustrate the wellhead, wellbore, and packer.
Response: Thank you for your comment. We have added sketches of wellbore, wellhead and packer structures to the manuscript, as shown in Figure 3.
Figure 3. Wellbore, wellbore and packer structure diagram
Q8: Why do you use these parameters? Why didn't you use the DOE to control the studied factors?
Response: Thank you for your comment. The experimental parameters in this paper are from the field data provided by the oilfield, from which we choose representative parameters for experiments. In fact, our main purpose is to take the corrosion of various pipes under the same working conditions as the basis for pipe selection. Therefore, we have selected such factors as phase state, total pressure, temperature, CO2 partial pressure and H2S partial pressure that have significant impact on corrosion, which are also the main factors for studying oil and gas field corrosion.
Q9: More illustration of this process is required to be added in the manuscript.
Response: Thank you for your comment. We have further added in the manuscript the detailed process of this step in the experimental process. The supplementary content is as follows:
100 mL hydrochloric acid (1.19 g/cm3), 900 mL distilled water and 10 g hexamethylenetetramine were prepared into a film removal solution (soak the test piece in the film removal solution and let it stand for a period of time. Wipe the test piece gently with a dust-free cloth to make the corrosion products on the surface of the test piece fall off), and the above samples were washed with the film removal solution to remove corrosion scales on the surface of the samples.
Q10: Reference of the used equation, fonts of the used symbols should be revised.
Response: Thank you for your careful suggestions and we have added equation reference and changed the fonts of all the symbols in the manuscript.
Q11: What is the name and model of the used device?
Response: Thank you for your comment and we have added the name and model of the used device in the manuscript. The revised content is as follows:
Three-dimensional microscope (3D optical microscope, Bruker ContourGT-K) is used to observe the surface morphology of the samples after removing the corrosion products and test the local corrosion depth.
Q12: How many samples did you test? Did you consider the statistical analysis of your experimental data?
Response: Thank you for your comment. Six test pieces were set for each group of experiments for each tube, three of which were used to calculate the corrosion rate, and the rest were used to observe the corrosion morphology. The analysis of experimental data is used to compare corrosion rates of different pipes and under different working conditions, so as to provide guidance for material selection at each position of acid gas reinjection well. Therefore, the comparative analysis of experimental data is mainly considered.
Q13: 0.076 mm/y, why do you use this reference value?
Response: Thank you for your comment. 0.076 mm/y is the basis for the evaluation of the corrosion degree of pipes in the oil and gas industry (SY/T 5329-2012 standard). It is generally believed that when the corrosion is less than 0.076, no protective measures need to be taken, otherwise, protective measures need to be taken. Therefore, we use this data as a reference for corrosion degree.
Q14: How do you prepare your cross-section for taken these images?
Response: Thank you for your comment. There is a complete set of operation sample for section analysis and sample preparation. In simple terms, A section sample with a thickness of 20mm wall was taken from the middle section of the tubing and sealed with epoxy resin. After the epoxy resin was air-dried, the sections needed to be observed were polished with water sandpaper of 360#, 600#, 800#, 1000# and 1200#, and the polished samples were placed in the scanning electron microscope to observe the corrosion section of the failed tubing.
Q15: Resolution needs to be enhanced.
Response: We have further enhanced the resolution of the XRD analysis in this paragraph. The enhanced content is as follows:
X-ray diffractometer was used to analyze the phase composition of the corrosion products of the corroded samples to obtain the XRD pattern. Jade software was used to compare and analyze the diffraction peak pattern obtained with the reference material card, so as to judge the phase composition of the corrosion products of the samples. The phase composition of corrosion scales of T95 steel in the liquid and gas phase are shown in Fig.15. XRD results demonstrated that the corrosion products of carbon steel are mackinawite and iron sulfide (FeS) and Ferrous carbonate (FeCO3).

Round 2
Reviewer 2 Report
I would like to thank the authors for their kind sufficient and valuable replies.